# Characterization of Structural, Optical, Corrosion, and Mechanical Properties of HfO_2_ Thin Films Deposited Using Pulsed DC Magnetron Sputtering

**DOI:** 10.3390/ma16145005

**Published:** 2023-07-14

**Authors:** Ewa Mańkowska, Michał Mazur, Małgorzata Kalisz, Marcin Grobelny, Jarosław Domaradzki, Damian Wojcieszak

**Affiliations:** 1Faculty of Electronics, Photonics and Microsystems, Wrocław University of Science and Technology, Janiszewskiego 11/17, 50-372 Wrocław, Poland; ewa.mankowska@pwr.edu.pl (E.M.); jaroslaw.domaradzki@pwr.edu.pl (J.D.); damian.wojcieszak@pwr.edu.pl (D.W.); 2Faculty of Engineering and Economics, Ignacy Mościcki University of Applied Sciences in Ciechanów, Narutowicza 9, 06-400 Ciechanów, Poland; malgorzata.kalisz@puzim.edu.pl; 3Faculty of Technical and Social Sciences, Ignacy Mościcki University of Applied Sciences in Ciechanów, Warszawska 52, 06-500 Mława, Poland; marcin.grobelny@puzim.edu.pl

**Keywords:** HfO_2_, nanohardness, Young’s modulus, corrosion resistance, optical properties, electrical properties, magnetron sputtering

## Abstract

Various properties of HfO_2_, such as hardness, corrosion, or electrical resistance, depend on the method and the conditions of deposition. In this work, a thorough comparison of scarcely investigated mechanical properties of HfO_2_ thin films deposited with different conditions of reactive magnetron sputtering process is presented. Four thin films were sputtered in processes that varied in plasma ignition method (continuous or sequential) and target–substrate distance. The structural characteristics of the HfO_2_ thin films were examined using Raman spectroscopy and X-ray diffraction measurements. Furthermore, the optoelectronic properties were determined based on transmittance and current–voltage characteristics. The mechanical properties of the HfO_2_ thin films were determined using nanoindentation and scratch test. In turn, the corrosion properties were determined by analyzing the voltametric curves. The transparent HfO_2_ thin films deposited in the continuous process are characterized by better corrosion resistance than the same layer formed in the sequential process, regardless of the target–substrate distance (8 cm or 12 cm). Furthermore, these samples are also characterized by the highest value of Young’s modulus and scratch resistance. The combination of good corrosion and scratch resistance could contribute to the new application of HfO_2_ as a corrosion protective material.

## 1. Introduction

Hafnium dioxide is one of the metal oxides that can be widely used in modern technologies. Its low optical absorption and dispersion, as well as high refractive index, make it an attractive material for many applications in the field of optoelectronics [1,2]. For example, HfO_2_-based thin films are used in accessories for high-power lasers, in cameras for space applications, as well as in antireflective coatings [3,4]. On the other hand, the high dielectric constant (k~25) and the wide energy gap (Eg~5.7 eV) [5,6] often result in the application of HfO_2_ as a dielectric material that replaces SiO_2_, for example, in transparent thin-film transistors as a gate material [7] or is used as an insulator in resistive free-access memory [8].

However, works on the mechanical properties of HfO_2_ thin films, such as wear resistance or corrosion resistance in various environments, including biological fluids are rather scarce [9]. It is worth paying more attention to the corrosion susceptibility of HfO_2_ because, as the Pourbaix diagram shows, hafnium is one of the most corrosion-resistant metals. This resistance is a result of the formation of the HfO_2_ layer, which is passive in a wide potential–pH range [10]. This oxide was considered in 1984 by Marcel Pourbaix as one of the materials for surgical implants [9]. Recently, oxides such as TiO_2_, ZrO_2_, Al_2_O_3,_ and HfO_2_ deposited by atomic layer deposition were investigated as protective barrier layers. Aluminum and hafnium oxides could be a protective layer of copper tubes in a 0.1 M NaCl solution. However, HfO_2_ provides better long-term protection [11,12]. Dinu [13] investigated the corrosion susceptibility of Ti(N,O)-coated stainless steel in a saline, acidic, and oxidizing environment. An additional coating of 10 nm of hafnium oxide increased corrosion robustness of stainless steel. The anticorrosion protection of HfO_2_ may also be important in photovoltaic applications. Staišiūnas [14] investigated the possibility of using amorphous hafnium(IV) oxide as a protective layer in the aggressive environments of silicon used in photovoltaic devices for water splitting. In an alkaline environment, it provided corrosion protection, but the layer was not irrelevant to the activity of the electrode. In acidic environments, the HfO_2_ film enhanced the effect of water splitting.

Furthermore, studies on the possibility of using HfO_2_ as protective coatings for implants were carried out for titanium alloys [9,15] and magnesium alloys such as AZ31 [16,17,18]. For such applications, the good thermal, chemical, and mechanical stability of hafnium oxide [1,2,3] and its biocompatibility and osteogenesis [9] are essential. The additional coating of HfO_2_ on magnesium or titanium alloys decreases the corrosion current density in the human body’s pH range (5.6 to 9) and improves cytotoxicity and cell viability [15,17]. However, most of the work published to date on corrosion properties concerns amorphous coatings deposited by ALD, while there is a lack of studies on crystalline films prepared using other methods, e.g., using magnetron sputtering. 

The mechanical properties of HfO_2_ coatings have so far been scarcely investigated. However, the determination of mechanical parameters of coatings, in particular for corrosion protective layers, is important. When two materials rub against each other, it can lead to abrasion of the corrosion protection layer. Tribocorrosion studies have shown that hafnium, after mechanical damage to a passive layer, has the ability to repassivate faster than titanium, but volume loss is greater [9].

However, the properties of HfO_2_ thin films strongly depend on the preparation method and deposition conditions. Mechanical properties and corrosion resistance strongly depend on the polymorphs of hafnium dioxide, i.e., monoclinic, tetragonal, and cubic. The most thermodynamically stable at room temperature and pressure is the monoclinic (m-HfO_2_) polymorph. The transformation of m-HfO_2_ into a tetragonal (t-HfO_2_) or cubic (c-HfO_2_) structure occurs at temperatures higher than 1995 K and 2695 K, respectively [19,20]. 

Magnetron sputtering provides the opportunity to tailor optical, electrical, and mechanical properties of the thin film by precisely adjusting the deposition parameters such as the type of plasma generation method, substrate temperature, applied power, and O_2_/Ar ratio. Zahoor [21] measured hardness values for polycrystalline HfO_2_ thin films prepared by magnetron sputtering in different O_2_/Ar ratios. At a lower O_2_/Ar ratio, a lower homogeneity of hardness in the layer was obtained. The hardness of such layers was approximately 12 GPa and the Young’s modulus increased with the increasing O_2_ flux, and the highest obtained hardness and Young’s modulus were 12.53 GPa and 208.08 GP, respectively. On the other hand, Vargas et al. [22] by changing the Ar to O_2_ ration obtained metallic, amorphous, and monoclinic HfO_2_ thin films of various mechanical parameters. The highest values of hardness and Young’s modulus were achieved for crystalline thin films (metallic Hf and monoclinic HfO_2_), and the values decreased with the amorphization of the film films. Mazur et al. [23] studied the effect of power on various properties of HfO_2_ thin films. As power increased, the hardness and modulus of elasticity increased, which could be due to the higher packing density and lower porosity of the films. Additionally, the refractive index increased with increasing power. 

To date, the influence of the parameters of the reactive magnetron sputtering process, such as position of the sample in the deposition chamber and the plasma-inducing method (continuous or sequential) on various properties of HfO_2_ thin films has never been studied. Therefore, in this paper, not only the influence of various deposition parameters on structural, optical, and electrical properties were described but also a deeper insight into mechanical and corrosion properties of monoclinic HfO_2_ thin films was presented. 

## 2. Materials and Methods

HfO_2_ thin films were deposited from high-purity (99.95%) metallic Hf targets using pulsed DC magnetron sputtering process. Coatings were sputtered in pure oxygen (working gas, e.g., argon, was not introduced into the chamber). The base pressure in the deposition chamber was equal to 5 × 10^−5^ mbar, while the sputtering processes were carried out with a pressure of 1.2 × 10^−2^ mbar. In each deposition process, sputtering conditions such as pressure, oxygen flow (18 sccm), and power (450 W) were maintained the same. Magnetron with Hf target was powered using DPS pulsed DC power suppliers working in the unipolar mode with 165 kHz sinusoidal pulses grouped at a frequency of 1.6 kHz with the voltage amplitude up to 1.8 kV. Four sets of HfO_2_ thin films were deposited due to the change in magnetron powering type and target–substrate distance. Coatings were deposited in processes with constant and intermittent powering of a magnetron. In the case of constant powering, magnetron was continuously supplied with voltage, while in intermittent powering, magnetron was supplied for 1 s with 1 s break. Therefore, these processes were called as continuous and sequential, respectively. Furthermore, the influence of a target–substrate distance (8 cm and 12 cm) on thin films properties was analyzed. Due to changes in magnetron powering and target–substrate distances, the sputtering process times were also adequately selected (Table 1). Hafnium dioxide coatings were deposited on unheated substrates of fused silica and TiAlV alloys in order to determine their microstructure, optical, mechanical, and electrochemical properties. Thin films deposited on fused silica were used for X-ray diffraction (XRD), Raman spectroscopy, transmission measurements, scratch tests, as well as optical imaging with the aid of an optical microscope and profilometer. For electrical measurements, corundum ceramic substrates with interdigitated platinum–gold electrodes were used, while thin films on Ti6Al4V substrates were destined for corrosion resistance and mechanical tests. Deposited thin film had a thickness in the range of 340 to 400 nm. Detailed information of deposition conditions are summarized in Table 1.

The structural properties of the prepared thin films were measured with the use of XRD and Raman spectroscopy. A PANalytical Empyrean PIXel3D powder diffractometer was equipped with a Cu Kα X-ray source (λ = 1.54056 Å). The Scherrer equation was used to estimate the crystallite size [24,25]. Raman spectra were measured using a Thermo Scientific, Waltham, MA, USA, DXR™ Raman Microscope. The spectra were recorded in the range from 90 to 800 cm^−1^ with a resolution of ca. 1 cm^−1^. The excitation source was a 455 nm blue laser diode at a power of 8 mW. During measurement, 10 scans were performed for each sample with an exposure time of 90 s.

Characterization of optical properties was performed by measuring transmittance in the wavelength range of 210 to 1000 nm. A coupled deuterium–halogen lamp was used as a light source, and the characteristics were obtained using an Optics QE65000 spectrophotometer. The analysis allowed to determine the average transmission in the visible wavelength range. Furthermore, the average transmittance at 550 nm was determined using the envelope method with Equation (1) [26]: (1)$$T_{i\;\left(\lambda =550\right)}=\frac{2\cdot T_{\lambda \;max}\;\cdot T_{\lambda \;min}}{T_{\lambda \;max}+T_{\lambda \;min}}$$
where *T_λmax_* is the transmission value of the envelope that passes through the maxima and *T_λmin_* is the transmission value of the envelope that passes through the minima.

On the basis of the results of transmission measurements, the absorption coefficient spectra were calculated using the following Equation (2):(2)$$a=\frac{1}{t}\cdot \ln\left(T\right)$$
where *t* is the thickness of the thin film. 

Using the reverse engineering method and FilmStar FTG v2.61.4405 software, the refractive index (*n*) and the extinction coefficient (*k*) were also calculated.

Electrical characterization was performed by measuring electrical resistivity at room temperature and at elevated temperature (up to 353 K) with the aid of a Keithley SCS4200 system and an M150 Cascade Microtech probe station.

Ti6Al4V titanium alloy was used as the substrate material for corrosion and nanoindentation tests. Before preparing the thin films, the titanium alloy surfaces were polished using the Stuers RotoPol 21 grinding and polishing apparatus. The surface of the samples was polished with emery paper and diamond suspension up to 0.05 µm to a ‘mirror image’. 

The corrosion behavior of titanium alloy and alloy with thin films was examined by an analysis of the voltametric curves, i.e., the extrapolation of the cathodic and anodic branch of the voltametric curves to the corrosion potential [27] in a three-electrode cell setup. In this setup, the titanium alloy and titanium alloy with thin films were used as working electrode, while a platinum and Ag/AgCl electrode with a Luggin capillary were used as counter and reference electrodes, respectively. Measurements were carried out in an electrolyte solution, which was composed of 0.5 M NaCl and 0.03 M KF and pH = 2 adjusted by concentrated hydrochloric acid with a scan rate of 1 mV/s within the range of −150 to 1000 mV versus open circuit potentials (OCP). Measurements were carried out using a Princeton Applied Research VersaSTAT 3 potentiostat/galvanostat with VersaStudio v2.63.3 software in aerated solutions at room temperature.

The hardness and Young’s modulus measurements of the thin films were performed using a CSM Instruments (Peseux, Switzerland) nanoindenter equipped with a Vickers diamond indenter. Hardness and Young’s modulus were calculated using the method proposed by Oliver and Pharr [28]. Several measurements were carried out for various depths of nanoindentation (from 80 to 700 nm). To measure the ‘film-only’ properties and minimize the impact of the substrate, a method of nanoindentation measurement approximation was implemented [29].

Scratch tests were performed with respect to the ASTM F735 standard [30] using the Taber Oscillating Abrasion Tester 6160. The effect of the scratch test was evaluated using an Olympus BX51 optical microscope and a TalySurf CCI Lite Taylor Hobson optical profilometer. 

## 3. Results

### 3.1. Structural Characterisation of the HfO_2_

The results of XRD measurements of HfO_2_ thin films deposited in various magnetron sputtering processes are shown in Figure 1. All deposited coatings were nanocrystalline with crystallite sizes from 6.3 nm to 8.4 nm and exhibited a monoclinic HfO_2_ structure [31,32]. XRD patterns are quite similar in the case of all thin films; however, detailed analysis showed some differences. First, thin films deposited in sequential process had smaller crystallite sizes of ca. 20% than those prepared in continuous sputtering. Moreover, changes in the target–substrate distance from 8 cm to 12 cm lead to a decrease in crystallite sizes of ca. 10%. Detailed XRD results of HfO_2_ thin films regarding their crystallite sizes, interplanar distances, and type of occurring stress are summarized in Table 2.

Raman spectroscopy was used for further analysis of the structural properties of deposited thin films since it is a more surface-sensitive method for the determination of crystallinity and microstructure. Hafnium dioxide with monoclinic structure has 18 active modes, 9A_g_ and 9B_g_ [33,34,35]. The peaks that are visible in Figure 2 were in good agreement with the reference values and indicated that the deposited coatings had a monoclinic HfO_2_ structure. Moreover, theses peaks can be assigned to both active modes, i.e., A_g_ (107, 133, 148, 384, 499, 580, 671 cm^−1^) and B_g_ (241, 332, 548, 639 cm^−1^). 

### 3.2. Optical and Electrical Properties of HfO_2_ Thin Films

As shown in Figure 3a, HfO_2_ thin films deposited magnetron sputtering were highly transparent in measured region. A sharp decrease in the transmission was observed for wavelengths shorter than 250 nm, showing increased absorption below this wavelength–absorption spectra are shown in Figure 3b. The average transmittance in the visible range was equal to 89%, while the transmission measured using the envelope method for a wavelength equal to 550 nm was equal to 88% for all HfO_2_ thin films. Only negligible changes in the refractive index of the deposited hafnia coatings were observed as *n* at λ = 550 nm was in the range of 1.75 to 1.79 (Figure 3c). In addition, the extinction coefficient of HfO_2_ thin films (Figure 3d) in the visible wavelength range was of the order of 10^−3^, demonstrating relatively good optical quality and low optical losses of the coatings. The type of generated plasma during magnetron sputtering, as well as the target–substrate distance, did not significantly influence the optical parameters of the hafnium dioxide thin films. 

The electrical resistivity of HfO_2_ was evaluated based on the linear current–voltage characteristics presented in Figure 4a. The resistivity values for HfO_2_ thin films deposited by sequential process was (7.9 ± 1.1) × 10^7^ Ωcm and were similar to the resistance of the thin film deposited by continuous process with a target–substrate distance equal to 8 cm. The resistivity of the film deposited in the continuous process with greater distance between the substrate and the target was over two times higher. In Figure 4b, the temperature-dependent resistivity plot is presented. On a semi-logarithmic scale, the relations of the resistivity and temperature in the range (303 ÷ 353 K) were linear. Thermally generated free charge carriers caused the resistance decrease. The activation energies (E_a_) were in the range from 0.19 eV/K to 0.31 eV/K and were evaluated according to Arrhenius Equation (3) [36,37]:(3)$$\rho =\rho_{0}\exp\left(\frac{E_{a}}{\mathrm{kT}}\right)$$
where k is the Boltzmann constant and T is the absolute temperature.

### 3.3. Corrosion Resistance and Mechanical Properties of HfO_2_ Thin Films

The electrochemical parameters of the HfO_2_ thin films deposited on Ti6Al4V substrates are summarized in Table 3. In turn, Figure 5 presents the voltametric curves of the titanium alloy and titanium alloy with HfO_2_ thin films. The calculated corrosion current density (i_corr_) was found to be equal to 3.63 × 10^−8^ A/cm^2^ and 1.26 × 10^−9^ A/cm^2^ for films deposited at the target–substrate distance of 8 cm during sequential and continuous processes, respectively. Changing the target–substrate distance to 12 cm resulted in obtaining corrosion current densities of 3.09 × 10^−8^ A/cm^2^ and 4.33 × 10^−9^ A/cm^2^ for films sputtered in sequential and continuous processes, respectively. These results show that a low corrosion rate and high electrochemical corrosion resistance were achieved by all HfO_2_ coatings on the titanium alloy surface. However, the smallest corrosion current density and therefore the best corrosion properties were obtained for HfO_2_ thin film deposited in continuous magnetron sputtering processes. Changes in target–substrate distance from 8 to 12 cm do not strongly affect the i_corr_ value. Only the influence of the magnetron powering, i.e., sequential or continuous, is noticeable. 

The hardness and Young’s modulus of the prepared HfO_2_ coatings were measured by a nanoindentation technique and determined using an approximation method. Additionally, the root mean square error (RMSE) was calculated. The results of the measurements of the mechanical parameters of the samples obtained from the approximation method are shown in Table 4. 

As results show, changing the distance between sample and target caused the change in hardness of obtained thin film, while maintaining the Young’s modulus constant. By increasing the target–substrate distance from 8 cm to 12 cm, the hardness of HfO_2_ thin film decreased, for sequential processes from 7.12 GPa to 6.45 GPa, while for continuous processes from 7.07 GPa to 6.27 GPa, respectively (see Figure 6a). When changing the type of deposition process, and maintaining a constant target–substrate distance, this time a change in the value of Young’s modulus was observed. The change in the type of magnetron powering method from sequential to continuous causes an increase in the value of Young modulus, from 93.3 GPa obtained for the sequential process carried out at a distance of 8 cm to 108.3 GPa for the continuous process carried out at the same distance. For the distance of 12 cm, the value of Young’s modulus increases from 88.8 GPa for sequential process to 108.0 GPa for continuous process (see Figure 6b). 

As demonstrated by Vargas et al. using magnetron sputtering with different ratios of Ar/O_2_ gas, it is possible to scale hardness values and Young’s modulus. The hardness ranged from 20 GPa for metallic hafnium to 9 GPa for the amorphous structure. The hardness for the well-crystallized monoclinic hafnium oxide was 15 GPa, and Young’s modulus was 164 GPa [22]. For CVD-deposited films, a hafnium silicide interlayer or a deliberately applied ZnO interlayer has been shown to increase the hardness of films with thickness below 100 nm [38,39].

As previous research has shown [22,40,41], it is possible to control mechanical parameters, such as Young’s modulus and hardness, of HfO_2_ thin films by changing their structure and chemical composition (content of Hf in the obtained thin film). Conducted research has shown that while the same structure of the HfO_2_ thin film (monoclinic) remains, the mechanical parameters of these coatings can be controlled by changing:-The type of magnetron powering from sequential to continuous to change the value of Young’s modulus;-The target–substrate distance to control the hardness.

Although the observed changes are subtle, they retain a certain consistency. This may prompt further research of the observed phenomenon for the obtained thin films.

To determine further the mechanical properties, scratch tests were performed. The thin films scratch resistance was evaluated based on images obtained using an optical microscope (Figure 7) and an optical profilometer (Figure 8). After tests, some scratches were visible on each thin film. Thin films deposited by the sequential process were less resistant to scratching, and the most noticeable scratches were obtained in the case of the film deposited with the target–substrate distance equal to 12 cm. Both thin films deposited in continuous process have a small number of scratches and with lower depths. The difference between the highest and lowest point on the surface was greater than 500 nm for the sequential process and less than 80 nm in the case of the continuous process. In each case, before conducting scratch tests, the surface was homogeneous and the roughness characterized with the Sq parameter was less than 1 nm. However, after performing scratch tests, the Sq parameter increased to 40 nm and 5 nm for thin films from sequential and continuous processes, respectively.

## 4. Summary

The subject of the work was focused on the HfO_2_ thin films deposited by magnetron sputtering. During the deposition, plasma was generated in continuous or sequential method, and the target–substrate distance in the sputtering chamber was equal to 8 cm or 12 cm. The premise of the work was to correlate the different deposition conditions with various properties of HfO_2_. To date, the optical and electrical properties of HfO_2_ thin films have been extensively investigated; however, the influence of the magnetron sputtering conditions on the mechanical and electrochemical properties still offers a plenty of space for exploration, e.g., for anticorrosive, scratch-resistant films for biomedical applications.

The corrosion and scratch tests clearly show that the HfO_2_ thin film deposited by continuous process is characterized by better corrosion and scratch resistance than the same layer formed during the sequential process, regardless of the position of the sample in the chamber (i.e., target–substrate distance of 8 cm or 12 cm). Mechanical characterization of obtained HfO_2_ thin films clearly shows that, by changing the parameters of the magnetron sputtering process, such as target–substrate distance and plasma-inducing method, it is possible to control its mechanical parameters, such as: hardness and Young’s modulus, and maintain the monoclinic structure of the obtained HfO_2_ thin films. Changing the distance between sample and target in the deposition chamber causes the change in hardness of obtained HfO_2_ thin film, while maintaining the Young’s modulus constant value. This phenomenon is completely independent of the method of inducing plasma during the sputtering method. Unlike the value of Young’s modulus that varies with change of the inducing plasma method and the position of the sample in the chamber. The observed phenomena require further thorough research. 

Obtained results show a clear correlation between the value of Young’s modulus, scratch resistance, and corrosion resistance of the HfO_2_ thin films. Thin films formed during the continuous deposition process are characterized by a higher value of Young’s modulus and the best corrosion resistance. To explain the observed phenomenon, it is necessary to conduct further detailed studies.

## Figures and Tables

**Figure 1 materials-16-05005-f001:**
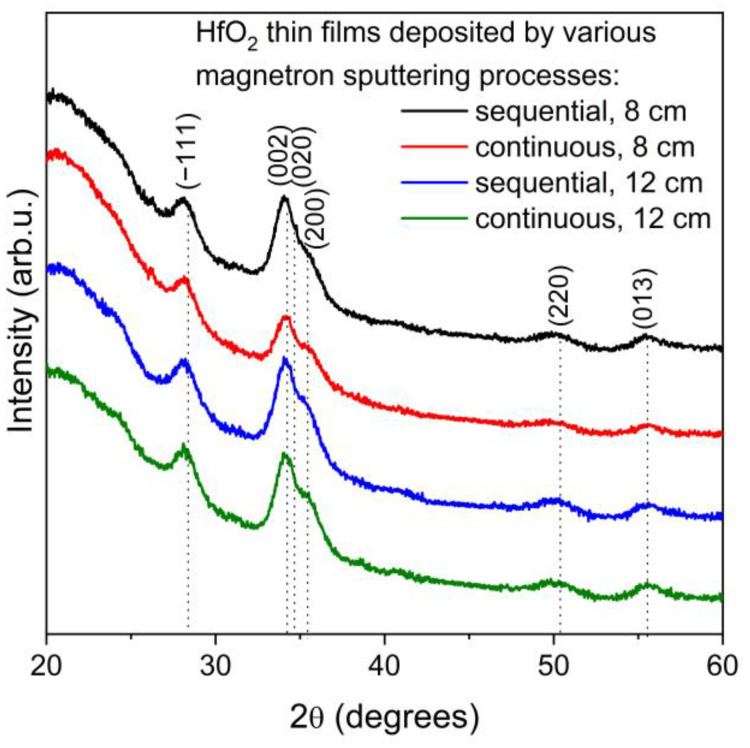
XRD diffraction patterns of HfO_2_ thin films deposited in various magnetron sputtering processes.

**Figure 2 materials-16-05005-f002:**
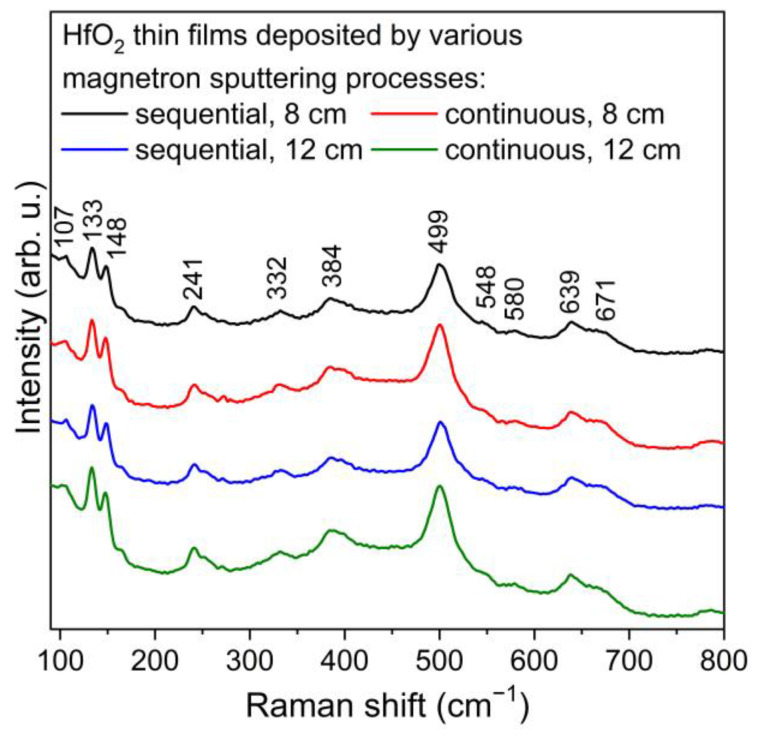
Raman spectra of HfO_2_ thin films deposited in various magnetron sputtering processes.

**Figure 3 materials-16-05005-f003:**
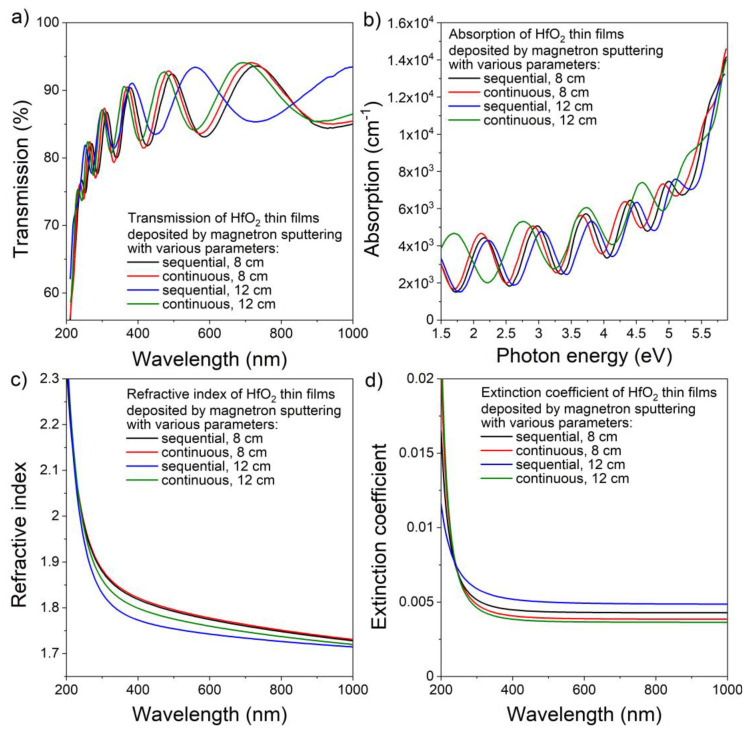
Results of the optical properties investigations of HfO_2_ thin films: (**a**) transmission and (**b**) absorption spectra, (**c**) refractive index, and (**d**) extinction coefficient.

**Figure 4 materials-16-05005-f004:**
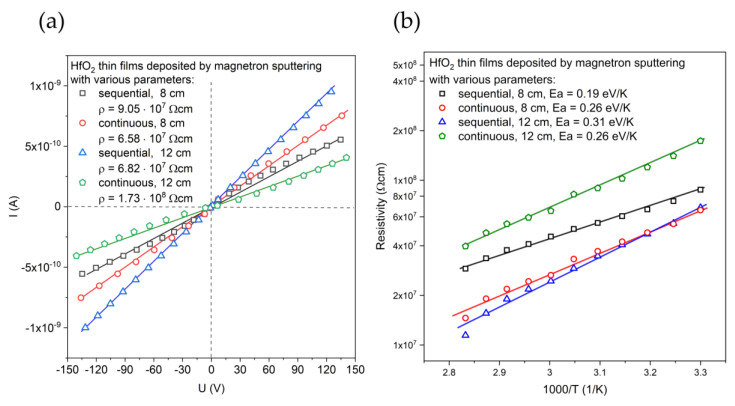
Results of electrical investigations: (**a**) current–voltage characteristics and (**b**) the resistivity of HfO_2_ thin films and its dependence on temperature. Symbols are measurement results, while solid lines are linear approximation.

**Figure 5 materials-16-05005-f005:**
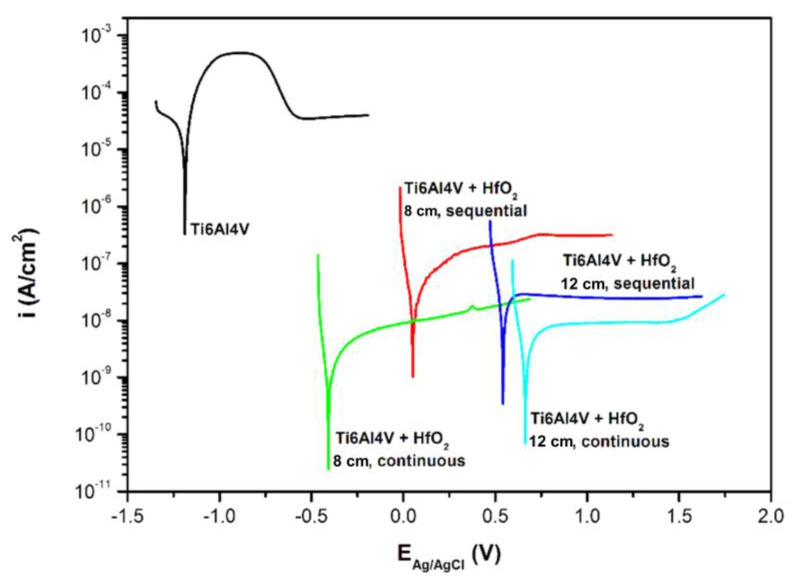
Voltametric curves of Ti6Al4V and titanium alloy with HfO_2_ thin films.

**Figure 6 materials-16-05005-f006:**
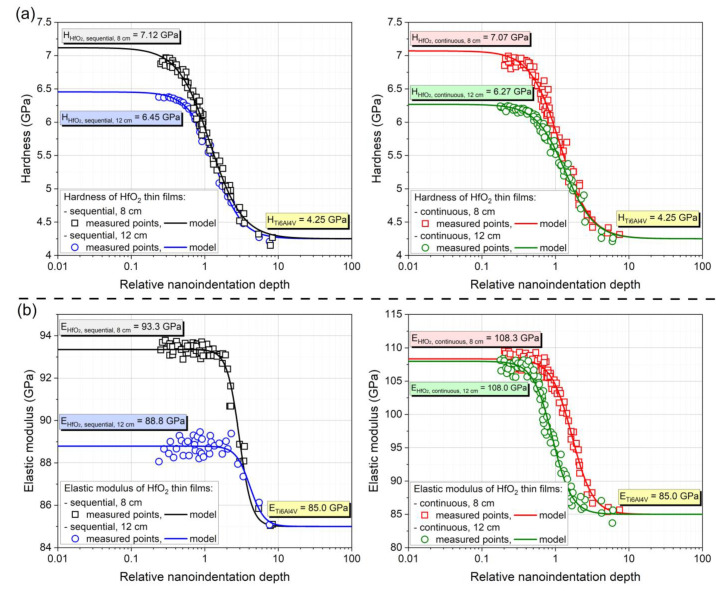
Results of (**a**) hardness and (**b**) Young’s modulus investigation for HfO_2_ thin films deposited at various target–substrate distances and magnetron powering method.

**Figure 7 materials-16-05005-f007:**
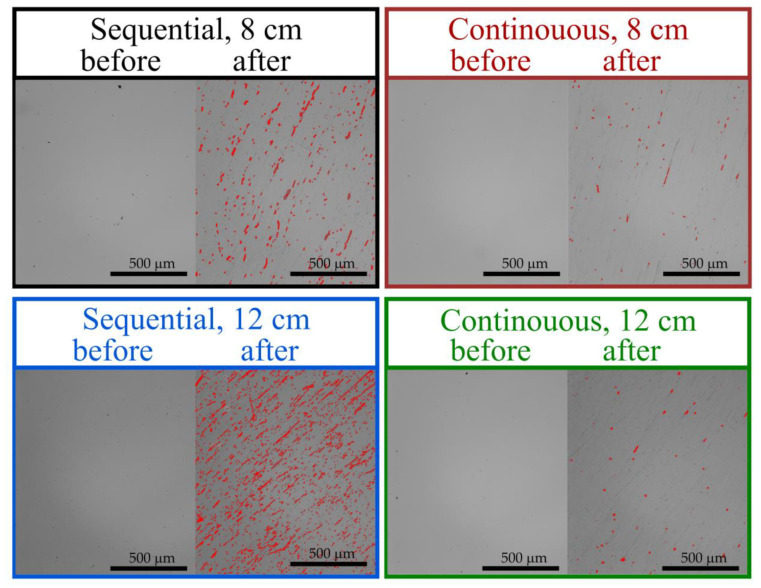
Images obtained using optical microscope of HfO_2_ thin films before and after scratch tests.

**Figure 8 materials-16-05005-f008:**
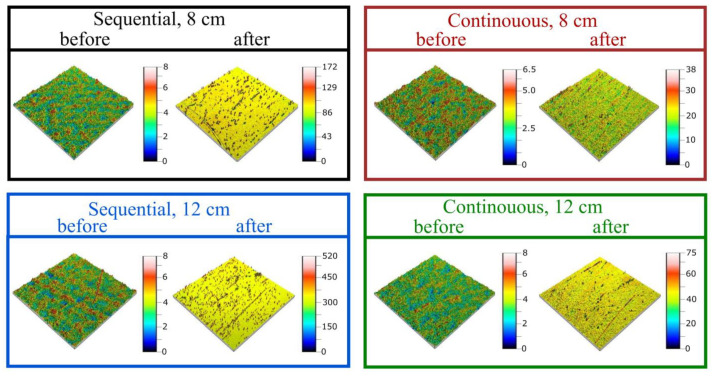
Images obtained using optical profilometer of HfO_2_ thin films before and after scratch tests.

**Table 1 materials-16-05005-t001:** Deposition conditions of HfO_2_ thin films.

MS Type	Target–Substrate Distance (cm)	Time of Powering Break (s)	Sputtering Time (min)	Power (W)	Sputtering Pressure (mbar)	Oxygen Flow (sccm)
sequential	8	1	180	450	1.2 × 10^−2^	18
	12		360			
continuous	8	-	90			
	12		180			

**Table 2 materials-16-05005-t002:** Results of XRD analysis performed for HfO_2_ thin films deposited in various magnetron sputtering processes.

Thin Film	MS Process	Target–Substrate Distance (cm)	D (nm)	d (nm)	d_PDF_ (nm)
m-HfO_2_	sequential	8	6.9	0.3160	0.3145
m-HfO_2_	continuous		8.4	0.3155	
m-HfO_2_	sequential	12	6.3	0.3159	
m-HfO_2_	continuous		7.5	0.3156	

Designations: D—average crystallites size, d—interplanar distance, d_PDF_—standard interplanar distance.

**Table 3 materials-16-05005-t003:** Electrochemical parameters of the titanium alloy and thin films on Ti6Al4V alloy.

Thin Film	MS Process	Target–Substrate Distance (cm)	i_corr_ (A/cm^2^)	E_corr_ (V)
HfO_2_	sequential	8	3.63 × 10^−8^	0.052
	continuous		1.26 × 10^−9^	−0.410
	sequential	12	3.09 × 10^−8^	0.542
	continuous		4.33 × 10^−9^	0.663
as-received sample Ti6Al4V			6.10 × 10^−5^	−1.188

**Table 4 materials-16-05005-t004:** Mechanical parameters of HfO_2_ thin films deposited on titanium alloy.

Thin Film	MS Process	Target–Substrate Distance (cm)	Hardness (GPa)	Elastic Modulus (GPa)
HfO_2_	Sequential	8	7.12 ± 0.12	93.3 ± 2.1
	Continuous		7.07 ± 0.09	108.3 ± 3.4
	Sequential	12	6.45 ± 0.10	88.8 ± 2.6
	Continuous		6.27 ± 0.10	108.0 ± 3.1

## Data Availability

Not applicable.

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
