# Peer review of "Characterization of Structural, Optical, Corrosion, and Mechanical Properties of HfO2 Thin Films Deposited Using Pulsed DC Magnetron Sputtering"

_materials, 2023, doi:10.3390/ma16145005_

Round 1

Reviewer 1 Report

In this work, the authors present various characterization techniques to study HfO2 thin films deposited by pulsed DC magnetron sputtering. However, the following points must be added before acceptance for publication.

1. x-ray should be rewritten as X-ray throughout the manuscript.

2. On page 2, line 65, change the wording "deficiency of studies" into something more appropriate like "no studies have been studied yet".

3. In the experimental part, mention the deposition time for preparation of different conditions of the samples.

4. Since you mentioned different substrates were used. Please emphasize in the results and discussions part which substrates were used for which measurement.

5. The authors claimed from XRD that their samples are nanocrystalline, however, the samples have been deposited at room temperature and the peaks are broad. Maybe it's amorphous in nature. Explain the reason behind this statement.

6. Is there any morphological differences observed among the different batches of the samples?

7. What is the thickness of your films? It must be added in the experimental section too.

8. Authors claimed that the Peak shift from XRD is due to the strain. How tensile stress is induced in your samples as there is no annealing treatment or substrate temperature provided to your sample?

9. Williamson-Hall (WH) plot must be used to calculate the type of strain existing in the samples.

10. Authors claimed from the Raman spectra that the change in the intensity is due to a higher degree of crystallization of the thin films. But XRD didn't reveal such a change. Moreover, this change could be due to the thickness of the samples that have been deposited over the substrate. Proper explanation or proof must be included for this claim or statement.

11. Did the authors deposit any top electrode for I-V measurements ?

12. The conclusion must be rewritten in a precise way with the importance of this work.

Author Response

Answers to the report of Reviewer #1

on the manuscript entitled: “Characterization of structural, optical, corrosion and mechanical properties of HfO2 thin films deposited by pulsed DC magnetron sputtering”

Authors: Ewa Mańkowska, Michał Mazur, Małgorzata Kalisz, Marcin Grobelny, Jarosław Domaradzki, Damian Wojcieszak

Authors:

We would like to express our gratitude for your remarks, which let us improve our manuscript. We have taken them into account in the revised version of our paper.

Answering to the Reviewer’s remarks, we have introduced some revisions in the manuscript.

Detailed response can be found in attached file.

Reviewer:

x-ray should be rewritten as X-ray throughout the manuscript

Authors:

Author are thankful for the comment – it was corrected.

Reviewer:

On page 2, line 65, change the wording "deficiency of studies" into something more appropriate like "no studies have been studied yet".

Authors:

The sentence was rewritten.

“However, most of the work published to date on corrosion properties concerns amorphous coatings deposited by ALD, while there is a lack of studies on crystalline films prepared by other methods, e.g. by magnetron sputtering.”

Reviewer:

In the experimental part, mention the deposition time for preparation of different conditions of the samples.

Authors:

The deposition time is included in the Table 1 of the manuscript. The experimental part was extended:

“Due to changes in magnetron powering and target-substrate distances the sputtering process times were also adequately selected (Table 1).”

Table 1. Deposition conditions of HfO2 thin films.

MS type

Target-

substrate

distance (cm)

Time of powering break (s)

Sputtering time (min)

Power (W)

Sputtering pressure (mbar)

Oxygen flow (sccm)

sequential

8

1

180

450

1.2·10-2

18

12

360

continuous

8

-

90

12

180

Reviewer:

Since you mentioned different substrates were used. Please emphasize in the results and discussions part which substrates were used for which measurement.

Authors:

In the experimental part, a detailed information concerning the substrate used for specific measurements was amended.

“Hafnium dioxide coatings were deposited on unheated substrates of fused silica, and TiAlV alloys in order to determine their microstructure, optical, mechanical, and electrochemical properties. Thin films deposited on fused silica were used for X-ray diffraction (XRD), Raman spectroscopy, transmission measurements, scratch tests, as well as optical imaging with the aid of optical microscope and profilometer. For electrical measurements corundum ceramic substrates with interdigitated platinum-gold electrodes were used, while thin films on Ti6Al4V substrates were destined for corrosion resistance and mechanical tests.”

Reviewer:

The authors claimed from XRD that their samples are nanocrystalline, however, the samples have been deposited at room temperature and the peaks are broad. Maybe it's amorphous in nature. Explain the reason behind this statement.

Authors:

Thin films were deposited on fused silica, which is an amorphous, highly transparent SiO2 glass. The amorphous HfO2 on silica substrate can be seen in the figure below. There are no diffraction peaks, even broad ones. In the case of nanocrystalline HfO2, the peaks are broad, which testifies about their small crystallite size of about several nanometers.

Although thin films were deposited without using additional heating or substrate bias, it was possible to obtain nanocrystalline coatings. It might be due to the highly energetic deposition process, sufficient to form crystalline thin films.

Fig. 1. Diffraction peaks of nanocrystalline and amorphous HfO2 deposited on fused silica

Reviewer:

Is there any morphological differences observed among the different batches of the samples?

Authors:

Two batches of samples were deposited and quick morphology investigation using optical profiler was done. There are only a slight differences in the morphology of the samples. In the case of thin film deposited in continuous process with the distance between substrate and magnetron equal to 8 cm in both batches the Sq parameter is close to 1. That testifies to homogeneous, flat morphology of the thin films.  

Table 1. The comparison of morphology and roughness of two batches of thin film deposited in continuous process with the distance between substrate and magnetron equal to 8 cm

Batch 1

Batch 2

Sq= 0.88

Sq=1.13

Reviewer:

What is the thickness of your films? It must be added in the experimental section too.

Authors:

The thickness of the thin films was in the range of 310 to 350 nm. It was added to the experimental section.

Reviewer:

Authors claimed that the Peak shift from XRD is due to the strain. How tensile stress is induced in your samples as there is no annealing treatment or substrate temperature provided to your sample?

Authors:

Using magnetron sputtering, it is possible to deposit layers with diverse properties by varying the deposition conditions. Magnetron sputtering is generally a method that uses high-energy particles, resulting in compressive and tensile stresses. One of the parameters affecting the type of stress is the sputtering pressure. An increase in sputtering pressure increases the number of scattering events, decreases the mean free path, and this leads to an increase in tensile stresses.

Reviewer:

Williamson-Hall (WH) plot must be used to calculate the type of strain existing in the samples.

Authors:

The average macrostrains existing in the samples were calculated using Williamson-Hall method.  As an example for sequential process with substrate-target position equal to 12 cm, the W-H plot is presented below. The calculated microstrain was equal to 0.018, but the R-square parameter was drastically low (0.57). This method seems to be irrelevant for calculating the microstrains in obtained HfO2 thin films, as the value is excessive big and the linear fitting in mediocre. In obtained diffractograms were observed only 6 reflection and no reflection from family of planes was found.

Reviewer:

Authors claimed from the Raman spectra that the change in the intensity is due to a higher degree of crystallization of the thin films. But XRD didn't reveal such a change. Moreover, this change could be due to the thickness of the samples that have been deposited over the substrate. Proper explanation or proof must be included for this claim or statement.

Authors:

In the case of thickness, it was very similar to all deposited coatings. However, taking into consideration the Reviewer comment, the statement about the degree of crystallization was removed from the manuscript.

Reviewer:

Did the authors deposit any top electrode for I-V measurements

Authors:

A I-V measurements were done for thin films deposited on a special kind of the substrate - corundum ceramic substrates with interdigitated platinum-gold electrodes. The thin films were deposited only in an active area. The contacts were mechanically masked during the deposition. Below are images of the substrate before the deposition of thin films and figure showing dimensions of the substrate provided by the manufacturer (BVT Technologies).

 a)

b)

Images of: a) the substrate and b) figure with the substrate dimensions

The experimental part was extended:

“Thin films deposited on fused silica were used for XRD, Ramman spectroscopy, trans-mission measurements, scratch testing, profiler and optical microscopy investigation. For electrical measurements corundum ceramic substrates with interdigitated platinum-gold electrodes were used, while thin films on Ti6Al4V substrates were used for corrosion resistance and mechanical tests.”

Reviewer:

 The conclusion must be rewritten in a precise way with the importance of this work.

Authors:

The summary was extended by the comment of the importance of the work.

“The subject of the work was focused on the HfO2 thin films deposited by magnetron sputtering. During the deposition, plasma was generated in continuous or sequential method, and the target-substrate distance in the sputtering chamber was equal to 8 cm or 12 cm. The premise of the work was to correlate the different deposition conditions with various HfO2 properties. To date, the optical and electrical properties of HfO2 thin films have been extensively investigated, however the influence of the magnetron sputtering conditions on the mechanical and electrochemical properties still offers a plenty of space for exploration e.g. for anticorrosive, scratch resistant films for biomedical applications.”

Reviewer 2 Report

The authors report on Characterization of structural, optical, corrosion and mechani-cal properties of HfO2 thin films deposited by pulsed DC magnetron sputtering. Though, it is an interesting work, however the article is worth required further revision and significant improvements as the novelty of this very hard work is completely missing. I have mentioned a very few of examples below, but these are not all, the full manuscript requires extensive revision. 

 1.      The significance of the work should be clearly stated. Abstract should be written clearly with a specific background and significance that attracts the reader of this field. 

2.      Introduction needs revision with more up to date citation and details about the background of the work as well as scientifically sound English. 

3.      Authors should be careful about sentence like "thin films deposited in various magnetron sputtering processes". 

4.      The results should be discussed with the comparison with other published similar works as H2F is not a significantly new materials to work. Such as in line 179-180 "Moreover, change of the target-substrate distance from 8 cm to 12 cm lead to a decrease of a crystallite sizes of ca. 10%." Did author compare their finding with other similar findings of same materials produced by different methods or sample prepared varying the deposition process parameters? 

5.      In lines 213-215 "As shown in Figure 3 thin films of HfO2 deposited magnetron sputtering were highly transparent in measured region. A sharp decrease in the transmission was observed for the wavelengths shorter than 250 nm. For longer wavelength interference fringes appeared. It is a common phenomenon that appears for thin films thicker than 100 nm." These statements are not strong argument for presenting results, most thin-films shows lower transmission below 300 nm and the longer wavelength interferences are not common phenomena for all types of transparent thin-films of above 100 nm thick. 

8.      The presented results should have proper explanation with solid evidence. 

Conclusions need to be written with solid explanation about the obtained results, findings and also the applications of this work.

Very poor

Author Response

Answers to the report of Reviewer #2

on the manuscript entitled: “Characterization of structural, optical, corrosion and mechanical properties of HfO2 thin films deposited by pulsed DC magnetron sputtering”

Authors: Ewa Mańkowska, Michał Mazur, Małgorzata Kalisz, Marcin Grobelny, Jarosław Domaradzki, Damian Wojcieszak

Authors:

We would like to express our gratitude for your remarks, which let us improve our manuscript. We have taken them into account in the revised version of our paper.

Answering to the Reviewer’s remarks, we have introduced some revisions in the manuscript.

Reviewer:

The significance of the work should be clearly stated. Abstract should be written clearly with a specific background and significance that attracts the reader of this field.

Authors:

The abstract was rewritten.

“Various properties of HfO2, such as hardness, corrosion, or electrical resistance, depend on the method and conditions of deposition. In this work, a thorough comparison of scarcely investigated mechanical properties of HfO2 thin films deposited with different conditions of reactive magnetron sputtering process was presented. Four thin films were sputtered in the processes that varied in plasma ignition method (continuous or sequential) and target-substrate distance. The structural characteristics of the HfO2 thin films were examined using Raman spectroscopy and X-ray diffraction measurements. Furthermore, the optoelectronic properties were determined based on transmittance and current-voltage characteristics. The mechanical properties of the HfO2 thin films were determined using nanoindentation and scratch test. In turn, the corrosion properties were determined by analyzing the voltammetric curves. The transparent HfO2 thin films deposited in the continuous process are characterized by better corrosion resistance than the same layer formed in the sequential process, regardless of the target-substrate distance (8 cm or 12 cm). Moreover, these samples are also characterized by the highest value of Young’s modulus and scratch resistance. Combination of good corrosion and scratch resistance might contribute to new application of HfO2 as corrosion protective material.”

Reviewer:

Introduction needs revision with more up to date citation and details about the background of the work as well as scientifically sound English.

Authors:

The introduction has been updated with literature published in the last 5 years. More background has been provided and the English language has been checked.

Reviewer:

Authors should be careful about sentence like "thin films deposited in various magnetron sputtering processes".

Authors:

In the descriptions of the figures the sentence was changed to more precise formulation “HfO2 thin films deposited by magnetron sputtering with various parameters”.

Reviewer:

The results should be discussed with the comparison with other published similar works as H2F is not a significantly new materials to work. Such as in line 179-180 "Moreover, change of the target-substrate distance from 8 cm to 12 cm lead to a decrease of a crystallite sizes of ca. 10%." Did author compare their finding with other similar findings of same materials produced by different methods or sample prepared varying the deposition process parameters?

Authors:

Of course, HfO2 is not a new material that has been deposited using magnetron sputtering. Other work on different powers applied to magnetrons during the deposition of HfO2 thin films has been carried out on the same workstation [22]. It was shown that increasing the power improves the crystallinity of the thin film and increases the size of the crystallites (nanocrystals). Another research group [X] studied the effect of the substrate-target distance on various properties of HfO2 thin films. The crystallinity of the thin films was similar at all tested distances, but the grains changed their shape from elliptical to circular as the substrate-target distance increased.

[22] Mazur, M.; Howind, T.; Gibson, D.; Kaczmarek, D.; Morgiel, J.; Wojcieszak, D.; Zhu, W.; Mazur, P. Modification of Various Properties of HfO2 Thin Films Obtained by Changing Magnetron Sputtering Conditions. Surf. Coatings Technol. 2017, 320, 426–431.

[X] Haque, S. M.; Rao, K. D.; Misal, J. S.; Tokas, R. B.; Shinde, D. D.; Ramana, J. V.; Rai, S.; Sahoo, N. K. Study of hafnium oxide thin films deposited by RF magnetron sputtering under glancing angle deposition at varying target to substrate distance. Appl. Surf. Sci. 2015, 353, 459-468.

Reviewer:

In lines 213-215 "As shown in Figure 3 thin films of HfO2 deposited magnetron sputtering were highly transparent in measured region. A sharp decrease in the transmission was observed for the wavelengths shorter than 250 nm. For longer wavelength interference fringes appeared. It is a common phenomenon that appears for thin films thicker than 100 nm." These statements are not strong argument for presenting results, most thin-films shows lower transmission below 300 nm and the longer wavelength interferences are not common phenomena for all types of transparent thin-films of above 100 nm thick.

Authors:

Transmission spectrum of a thin film can exhibit successive local maxima and minima (interference fringes) if several conditions are fulfilled:

-   thin film thickness is uniform – the interference effect gives rise to the particular part of the transmission spectrum showing some peaks (maxima) and valleys (minima)

-   thin film is deposited on a transparent substrates – for example fused silica;

-   thin film thickness is a lot smaller, i.e. several orders of magnitude, than that of the substrate – in our case the thin film thickness is in the range between the 300 and 350 nm and the thickness of the substrate is 0.5 mm;

In the case of our article, HfO2 thin films were thinner than the substrate and had higher refractive index than the fused silica, in the end showing interference fringes in the transmission spectrum.

According to the Reviewer remark a corresponding part of the manuscript was corrected.

“As shown in Figure 3 HfO2 thin films deposited by magnetron sputtering were highly transparent in the measured region. A sharp decrease in the transmission was observed for the wavelengths shorter than 250 nm showing increased absorption below this wavelength.”

Reviewer:

The presented results should have proper explanation with solid evidence.

Conclusions need to be written with solid explanation about the obtained results, findings and also the applications of this work.

Authors:

The summary was extended by the comment of the importance of the work.

“The subject of the work was focused on the HfO2 thin films deposited by magnetron sputtering. During the deposition, plasma was generated in continuous or sequential method, and the target-substrate distance in the sputtering chamber was equal to 8 cm or 12 cm. The premise of the work was to correlate the different deposition conditions with various HfO2 properties. To date, the optical and electrical properties of HfO2 thin films have been extensively investigated, however the influence of the magnetron sputtering conditions on the mechanical and electrochemical properties still offers a plenty of space for exploration e.g. for anticorrosive, scratch resistant films for biomedical applications.”

The obtained results were compared with other works:

“As demonstrated by Vargas et al. using magnetron sputtering with different ratios of Ar/O2 gas, it is possible to scale hardness values and Young's modulus. Hardness ranged from 20 GPa for metallic hafnium to 9 GPa for the amorphous structure. The hardness for the well-crystallised monoclinic hafnium oxide was 15 GPa and the Young's modulus was 164 GPa [22]. For CVD-deposited films, a hafnium silicide interlayer or a deliberately applied ZnO interlayer has been shown to increase the hardness of films with thicknesses of below 100 nm [40, 41].”

Reviewer 3 Report

It is necessary to give an explanation of the abbreviations Hit and Eit in Table 5 on page 8.

Author Response

Answers to the report of Reviewer #3

on the manuscript entitled: “Characterization of structural, optical, corrosion and mechanical properties of HfO2 thin films deposited by pulsed DC magnetron sputtering”

Authors: Ewa Mańkowska, Michał Mazur, Małgorzata Kalisz, Marcin Grobelny, Jarosław Domaradzki, Damian Wojcieszak

Authors:

We would like to express our gratitude for your remarks, which let us improve our manuscript. We have taken them into account in the revised version of our paper.

Answering to the Reviewer’s remarks, we have introduced some revisions in the manuscript.

Reviewer:

It is necessary to give an explanation of the abbreviations Hit and Eit in Table 5 on page 8.

Authors:

Authors are thankful for the remark. The Hit (meaning indentation hardness) was changed to Hardness and Eit (meaning indentation elastic modulus) was changed to Elastic modulus.

Round 2

Reviewer 2 Report

To complete the optical properties characterization add the optical constants of HfO2 and the wavelength dependent absorption coefficient spectra.

Author Response

Answers to the report of Reviewer #2

on the manuscript entitled: “Characterization of structural, optical, corrosion and mechanical properties of HfO2 thin films deposited by pulsed DC magnetron sputtering”

Authors: Ewa Mańkowska, Michał Mazur, Małgorzata Kalisz, Marcin Grobelny, Jarosław Domaradzki, Damian Wojcieszak

Authors:

We would like to express our gratitude for your remarks, which let us improve our manuscript. We have taken them into account in the revised version of our paper.

Answering to the Reviewer’s remarks, we have introduced some revisions in the manuscript.

Reviewer:

To complete the optical properties characterization add the optical constants of HfO2 and the wavelength dependent absorption coefficient spectra.

Authors:

Optical constants, such as refractive index and extinction coefficient, as also absorption coefficient spectra were added to the article. Materials and Methods and Results parts were extended with needed comments:

Materials and Methods part:

“The characterization of optical properties was performed by measuring transmittance in the wavelength range of 210 to 1000 nm. A coupled deuterium-halogen lamp was used as a light source, and the characteristics were obtained using an Optics QE65000 spectrophotometer. The analysis allowed to determine the average transmission in the visible wavelength range. Furthermore, the average transmittance at 550 nm was determined using the envelope method with equation 1 [26]:

                                                  (1)

where: Tlmax is the transmission value of the envelope that passes through the maxima and Tlmin is the transmission value of the envelope that passes through the minima.

On the basis of the results of transmission measurements, the absorption coefficient spectra were calculated using the following equation 2:

                                                                       (2)

where: t is the thickness of the thin film.

Using the reverse engineering method and FilmStar FTG software, the refractive index (n) and the extinction coefficient (k) were also calculated.”

Results part:

As shown in Figure 3a, HfO2 thin films deposited magnetron sputtering were highly transparent in measured region. A sharp decrease in the transmission was observed for wavelengths shorter than 250 nm, showing increased absorption below this wavelength - absorption spectra are shown in Fig. 3b. The average transmittance in the visible range was equal to 89% while the transmission measured using the envelope method for a wavelength equal to 550 nm was equal to 88% for all HfO2 thin films. Only negligible changes in the refractive index of the deposited hafnia coatings were observed as n at l=550 nm was in the range of 1.75 to 1.79 (Fig. 3c). In addition, the extinction coefficient of HfO2 thin films (Fig. 3d) in the visible wavelength range was of the order of 10-3, demonstrating relatively good optical quality and low optical losses of the coatings. The type of generated plasma during magnetron sputtering, as well as the target-substrate distance, did not influence significantly the optical parameters of the hafnium dioxide thin films.

Figure 3. Results of the optical properties investigations of HfO2 thin films: a) transmission and b) absorption spectra, c) refractive index and d) extinction coefficient.
